# The Ameliorative Effect of Pioglitazone against Neuroinflammation Caused by Doxorubicin in Rats

**DOI:** 10.3390/molecules28124775

**Published:** 2023-06-15

**Authors:** May M. Alsaud, Ahmad H. Alhowail, Maha A. Aldubayan, Ibtesam S. Almami

**Affiliations:** 1Department of Pharmacology and Toxicology, College of Pharmacy, Qassim University, Buraydah 51452, Al Qassim, Saudi Arabia; 431214396@qu.edu.sa (M.M.A.); m.aldubayan@qu.edu.sa (M.A.A.); 2Department of Biology, College of Science, Qassim University, Buraydah 51452, Al Qassim, Saudi Arabia; i.almami@qu.edu.sa

**Keywords:** DOX, PIO, cognitive impairment, chemotherapy, neuroinflammation

## Abstract

Doxorubicin (DOX) is a chemotherapeutic agent that is linked with complications such as cardiotoxicity and cognitive dysfunction, known as chemobrain. Chemobrain affects up to 75% of cancer survivors, and there are no known therapeutic options for its treatment. This study aimed to determine the protective effect of pioglitazone (PIO) against DOX-induced cognitive impairment. Forty Wistar female rats were equally divided into four groups: control, DOX-treated, PIO-treated, and DOX + PIO-treated. DOX was administered at a dose of 5 mg/kg, i.p., twice a week for two weeks (cumulative dose, 20 mg/kg). PIO was dissolved in drinking water at a concentration of 2 mg/kg in the PIO and DOX-PIO groups. The survival rates, change in body weight, and behavioral assessment were performed using Y-maze, novel object recognition (NOR), and elevated plus maze (EPM), followed by estimation of neuroinflammatory cytokines IL-6, IL-1β, and TNF-α in brain homogenate and RT-PCR of a brain sample. Our results showed a survival rate of 40% and 65% in the DOX and DOX + PIO groups, respectively, compared with a 100% survival rate in the control and PIO treatment groups at the end of day 14. There was an insignificant increase in body weight in the PIO group and a significant reduction in the DOX and DOX + PIO groups as compared with the control groups. DOX-treated animals exhibited impairment of cognitive function, and the combination PIO showed reversal of DOX-induced cognitive impairment. This was evidenced by changes in IL-1β, TNF-α, and IL-6 levels and also by mRNA expression of TNF- α, and IL-6. In conclusion, PIO treatment produced a reversal of DOX-induced memory impairment by alleviating neuronal inflammation by modulating the expression of inflammatory cytokines.

## 1. Introduction

Cancer survival rates have significantly improved over the years as a result of progress in cancer awareness, screening, prevention, diagnosis, and treatment. Chemotherapy is one of the most effective conventional treatments for patients with cancer [1], but survivors frequently experience cognitive disturbances following treatment with chemotherapy that include memory impairment associated with hippocampal dysfunction [2]. During and shortly after chemotherapy for cancer, many patients report attention deficits, memory loss, and confused thought processes [3]. Along with the increase in the number of effective chemotherapeutics that provide complete cure or satisfactory long-term control of the disease, there has been a rapid increase in the number of patients with “chemobrain” or “chemofog” [4,5]. Chemobrain, or chemofog, refers to the chemotherapy-associated patient condition that is characterized by a decline in cognitive functions such as attention, language, thinking, learning, memory, and problem-solving [6], and it is associated with a reduced quality of life among survivors [6]. Not all cancer patients develop chemobrain following chemotherapy, but the incidence of chemobrain is as high as 75% among patients with breast cancer and may persist in 17% to 34% of the patients [7]. These cognitive impairments experienced by the patients vary in range from mild to serious memory impairment [8]. Moreover, the duration of chemobrain symptoms ranges from short to long [9,10], with around one-third of the patients reporting side effects that last from several months to as long as 10 years after complete cessation of treatment [3,11]. Chemobrain is associated with neuroinflammation, mitochondrial dysfunction, and oxidative stress-related damage, but the complete mechanism has not been fully elucidated yet [2,12].

Doxorubicin (DOX) is commonly used in chemotherapeutic regimens to treat various solid and hematological malignancies [13]. It belongs to the class of anthracyclines that are known to exert their antitumor effects through DNA insertion and inhibition of topoisomerase II; in addition, DOX causes the production of invasive systemic reactive oxygen species [14]. DOX has limited ability to penetrate the blood–brain barrier (BBB), and as a result, the brain is largely protected from damage caused by DOX. However, despite its limited passage through the BBB, DOX can still cause severe neurotoxicity. Several clinical studies have reported cognitive impairment based on the results of cognitive ability tests in patients of all ages treated with DOX [15,16]. Variations in the volume and density of white matter and gray matter determined by MRI in patients undergoing chemotherapy [17] have revealed that the structure and function of the hippocampus are disrupted and neurogenesis is impaired, thus leading to cognitive impairment [18]. In addition, neuroinflammation plays a crucial role in the pathogenesis of brain diseases [19] and is known to be involved in the development of neurodegenerative diseases and cognitive decline [20]. Further, DOX has been found to induce peripheral inflammation by TNF-α generation that leads to central neuroinflammation, which is an important contributor to chemobrain induction [21]. Thus, neuroinflammation may be a potential treatment target for resolving the cognitive impairment caused by chemotherapy.

The peroxisome proliferator-activated receptor γ (PPARγ) receptor is distributed widely in the brain [22,23] and is crucial for learning [24]. In addition, PPAR-γ was found to be involved in regulating both cell survival and inflammatory responses and was abundant in neurons and astrocytes [25]. Pioglitazone (PIO) is a thiazolidinedione that acts as a PPAR-γ activator. It has gained interest due to its anti-inflammatory [26,27] and antioxidant properties [28] in neurodegenerative disorders. Recent studies have shown that specific PIO formulations can cross the BBB and cause subsequent improvements in the response of insulin receptors to insulin, regulate glucose metabolism in the brain, and reduce neuroinflammation [29,30]. Intriguingly, administration of PIO ameliorates cognitive dysfunction and increases the survival of dopaminergic neurons in animal models of Alzheimer’s disease and Parkinson’s disease [31,32]. Further, a recent randomized, open-controlled trial of patients with type 2 diabetes mellitus and mild Alzheimer’s disease reported that PIO treatment resulted in cognitive and functional improvements and reduced fasting plasma insulin levels [33]. With regard to the related mechanisms, PIO has been shown to ameliorate inflammation, oxidative stress, amyloidogenesis, and hypothyroidism, as well as enhance neurogenesis, synaptic plasticity, and mitochondrial function via its effects on relevant PPAR-mediated pathways [34,35]. Further, PIO can enhance the expression of proteins that are important for mitochondrial function and inhibit the increase in the production of pro-inflammatory cytokines such as IL-1, IL-6, and TNF-α [36]. These findings imply that PIO may have potential benefits for alleviating the cognitive impairment associated with DOX treatment. However, its effects on cancer patients with cognitive impairment after chemotherapy with DOX have not been explored yet. Therefore, this study aimed to test the hypothesis in experimental animals that DOX-induced cognitive impairment through elevation of neuroinflammation via increase in pro-inflammatory cytokines, such as TNF-α, IL-6, and IL-1β, can be reversed by co-administration of PIO with DOX. Despite the high prevalence of chemobrain in breast cancer patients and DOX being one of the most common antibiotics for breast cancer regimens, there has been limited research investigating the effect of DOX on the memory function of female rats and possible interventions. Therefore, the current study investigated the memory dysfunction in female rats following treatment with DOX and the possible ameliorative effect of PIO on cognition and neuroinflammation that could shed light on the development of new therapeutic interventions for the prevention and treatment of chemobrain. 

## 2. Results

### 2.1. Effect of DOX and PIO on Mortality

DOX treatment was found to affect the survival rate. DOX treatment produced a decline in survival rate in a duration-dependent manner and resulted in a 40% survival rate at the end of the two-week treatment schedule. However, the combination of PIO with DOX (DOX + PIO) resulted in an increase in survival rate (65%), whereas the animals in the control and PIO treatments endured 100% survival rates after two weeks of treatment (Figure 1).

### 2.2. Effect of DOX and PIO on Body Weight

As shown in Figure 2, the animals in the control and PIO-treated groups showed an increase in body weight, though the percentage change in body weight was statistically insignificant after the 2-week period compared with the first day of the study. However, there was a reduction in body weight in the DOX and DOX + PIO groups, and the percentage change in their body weight was statistically significant as compared with their corresponding weight on the first day of the study.

### 2.3. Effect of DOX and PIO on Performance in the Y-Maze Test

DOX treatment resulted in a significant (*p* < 0.05) reduction in the number of novel arm entries and a reduction in the duration of time spent on the novel arm, though statistically insignificant as compared with the control. There was a reversal of the DOX-induced reduction in the number of novel arm entries and time spent on the novel arm following treatment with PIO (DOX + PIO) as compared with DOX-treated animals (Figure 3a,b). There was no significant change in the number of novel arm entries or the duration of time spent on the novel arm following PIO treatment alone as compared with the control. Further, there was a reversal of the DOX-induced reduction in total arm entries by PIO (DOX + PIO), whereas PIO treatment alone did not produce significant changes in total arm entries as compared with control (Figure 3c).

### 2.4. Effects of DOX and PIO on Performance in the NOR Test

Figure 4 summarizes the effect of PIO on DOX-induced behavioral changes on the NOR test in rats. DOX treatment produced a significant (*p* < 0.05) reduction in time spent exploring the novel object as compared with the control. Concurrent treatment with PIO (DOX + PIO) produced an insignificant reversal of the DOX-induced decline in time spent exploring the novel object as compared with the DOX-treated group. However, PIO treatment also showed a reduction in time spent exploring the novel object, though statistically insignificant as compared with the control.

### 2.5. Effects of DOX and PIO on Performance in the EPM Test

Chronic treatment with DOX produced a significant increase in time spent in the closed arm and a significant reduction in time spent in the open arm of the EMP test as compared with the control (Figure 5a,b). Concurrent PIO treatment (DOX + PIO) produced a reduction in time spent in the closed arm and an increase in time spent in the open arms as compared with DOX, though statistically insignificant. PIO treatment alone did not produce significant changes in either time spent in the closed arms or time spent in the open arms as compared with the control (Figure 5a,b).

### 2.6. Effect of DOX and PIO on IL-1β Level in Rat Brain

Figure 6a represents the effect of DOX and PIO on IL-1β levels in the brain. DOX treatment for two weeks produced a significant increase in the brain’s IL-1β level as compared with the control, whereas PIO treatment alone did not produce significant changes in the IL-1β level as compared with the control. Concurrent treatment with PIO (DOX + PIO) insignificantly reduced the DOX-induced increase in IL-1β level as compared with the DOX-treated group.

### 2.7. Effect of DOX and PIO on TNF-α Level in Rat Brain

As shown in Figure 6b, DOX treatment significantly (*p* < 0.05) increased the brain’s TNF-α levels as compared with the control. Treatment with PIO (DOX + PIO) produced a significant (*p* < 0.0001) reduction in TNF-α levels as compared with the DOX-treated group. However, PIO treatment alone did not produce significant changes in brain TNF-α levels as compared with the control group.

### 2.8. Effect of DOX and PIO on IL-6 Level in Rat Brain

Chronic treatment with DOX for two weeks produced a significant (*p* < 0.05) increase in brain IL-6 levels as compared with the control. However, concurrent PIO treatment (DOX + PIO) produced a significant (*p* < 0.0001) reduction in IL-6 levels as compared with the DOX-treated group. PIO treatment alone also produced a significant (*p* < 0.05) reduction in IL-6 levels as compared with the control (Figure 6c).

### 2.9. Effect of DOX and PIO on mRNA Expression of IL-1β

As shown in Figure 7a, there was a significant (*p* < 0.0001) increase in the relative expression of IL-1β in the DOX-treated group as compared with the control group. The combination of PIO with DOX (DOX + PIO) produced a significant (*p* < 0.0001) reversal of the DOX-induced increase in the relative expression of IL-1β as compared with the DOX-treated group. However, PIO treatment alone produced a significant (*p* < 0.05) decrease in the relative expression of IL-1β as compared with the control group.

### 2.10. Effect of DOX and PIO on mRNA Expression of TNF-α

Figure 7b summarizes the findings of the RT-PCR analysis. DOX treatment produced a significant (*p* < 0.0001) increase in the relative expression of TNF-α as compared with the control group. Concurrent PIO treatment (DOX + PIO) resulted in a significant reduction in the DOX-induced increase in the relative expression of TNF-α as compared with the DOX-treated group. However, PIO treatment alone showed an insignificant (*p* > 0.05) elevation in the relative expression of TNF-α as compared with the control group.

### 2.11. Effect of DOX and PIO on mRNA Expression of IL-6

There was a statistically significant (*p* < 0.0001) elevation of IL-6 expression in the DOX-treated group as compared with the control. Concurrent PIO treatment (DOX + PIO) produced a significant reduction in the relative expression of IL-6 levels as compared with the DOX-treated group. However, PIO treatment alone also produced a significant (*p* < 0.05) reduction in the relative expression of IL-6 as compared with the control (Figure 7c).

**Figure 7 molecules-28-04775-f007:**
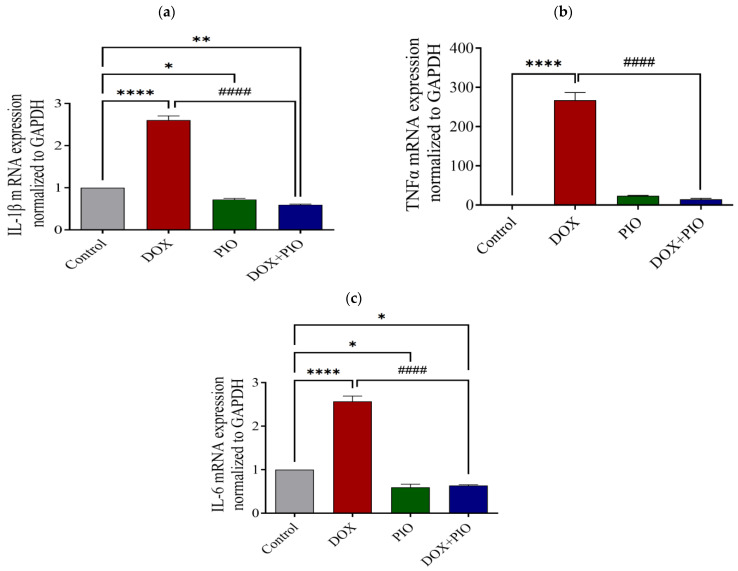
Effect of DOX and PIO treatment on relative expression of IL-1β (**a**), TNF-α mRNA (**b**), and IL-6 (**c**) in the rat brain. Data represented as mean ± SEM and analyzed by ANOVA followed by Tukey test. A *p* value < 0.05 was considered significant. * *p* < 0.05, ** *p* < 0.01, **** *p* < 0.0001 when compared with control and #### *p* < 0.0001 when compared with DOX.

## 3. Materials and Methods

### 3.1. Chemicals

DOX injection (2 mg/mL) was obtained from EBEWE Pharma Ges.m.b.H. Nfg.KG, Unterach am Attersee, Austria, and pioglitazone hydrochloride (Glados^®^) was obtained from Tabuk Pharmaceutical Manufacturing Co., Tabuk, Saudi Arabia.

### 3.2. Animals

Wistar female rats, 8–12 weeks old, weighing 150–250 g, were procured from the animal house facility of College of Pharmacy, Qassim University, Saudi Arabia. The animals were housed in propylene cages (5 animals/cage) under standard laboratory conditions (12 h light-dark cycle), maintained at 25 ± 2 °C, with free access to animal feed and water ad libitum. The animals were observed daily to determine their survival rate, and their body weights were measured every 3 days. All the behavioral tests were performed during the light phase of the cycle.

### 3.3. Drug Treatment

The animals were divided into four groups of 10 each: control, DOX, PIO, and DOX + PIO. DOX was administered intraperitoneally at a dose of 5 mg/kg twice a week for 2 weeks (total dose: 20 mg/kg). Glados (Pioglitazone hydrochloride, 30 mg) tablets were crushed and dissolved in aliquots of drinking water to obtain a concentration of 1 mg/mL and administered at a dose of 2 mg/kg by oral gavage needle once daily, starting on the same day of DOX treatment. The rats underwent behavioral tests after the completion of drug treatment, and then the animals were euthanized under ether anesthesia, and the brain was dissected promptly for estimation of proinflammatory cytokine levels in brain homogenate and RT-PCR of the brain. 

### 3.4. Y-Maze Test

The Y-maze test is a hippocampal-dependent task used to test spatial working memory. Here, spontaneous alternation behavior was assessed using a Y maze that was composed of three equally spaced arms composed of wood that were placed at an angle of 120° to each other and measured 50 × 10 × 18 cm each. The arms were painted brown to ensure easy visualization, and the apparatus was placed on the floor. Light was provided from above to ensure equal light distribution. This setup is conducive to spontaneous alternation behavior since rodents have a natural tendency to explore unfamiliar spaces. That is, when they face the option of entering two arms, they usually choose the one that has not been recently visited [37]. This gives the animal the ability to remember which arm has been visited a fewer number of times recently, and therefore, this test is used to assess working memory, which is a form of short-term memory [37,38]. In the training session, the rats were allowed to freely explore two arms for 15 min. During the second session (3 min duration), the animals were allowed to explore the entire maze, including the novel arm. The time interval between the two sessions was 3 h. The test sessions were video-recorded to determine the time spent by the animals in each arm and the number of entries (note: an animal was considered to have entered an arm even if half of its body was inside).

### 3.5. Novel Object Recognition Test

Ennaceur and Delacour developed the novel object recognition (NOR) test based on the knowledge that rats spend more time exploring a novel stimulus than a familiar stimulus [39]. The NOR test is based on a non-rewarded, relevant paradigm that uses the spontaneous exploratory behavior of rodents to measure recognition memory. In the most commonly used forms of the test, each test session consists of two trials. In the first trial (acquisition), the animals were exposed to two identical objects and allowed to freely explore the apparatus. During the second trial (retention), they were exposed to two dissimilar objects—one familiar object from the first trial and a new object. Object recognition can be assessed as the difference in time spent by the rodent exploring familiar and novel objects. In general, rodents have been found to spend more time exploring novel objects than familiar objects. The test apparatus was a wooden box (dimensions: 40 × 40 × 40 cm) with an open top. The familiarization objects were two black teacups, and the novel object was a black/white bottle (of equal size to the teacup). In the training session, the rats were allowed to explore the two teacups for 10 min and then return to their cages. In the second session conducted 3 h later with a duration of 3 min, one of the teacups was replaced with the novel object, and the time spent exploring the novel object was recorded using a video camera and a stopwatch.

### 3.6. Elevated plus Maze

Elevated plus maze (EPM) is a useful pharmacologically and ethologically validated tool for the assessment of rodent anxiety-like behavior [40,41]. It was described by Handley and Mithani as a test for the assessment of the anxiety behavior of rodents based on the ratio of time spent in the open arms to time spent in the closed arms. The EPM test relies on rodents’ tendency to move towards dark, enclosed spaces and an unconditioned fear of heights and open spaces. The EPM test uses a plus-shaped apparatus with four arms that are at right angles to each other [42]. The wooden apparatus consists of two opposing arms: the open arms (50 × 10 cm) and the closed arms (50 × 10 cm). The height of the sidewalls of the closed arms is 30 cm. The central platform between the arms measures 10 cm^2^. The maze was placed 50 cm above the floor. During the experiment, the rat was placed at the end of an open arm, facing the central platform, and allowed to explore the apparatus for 5 min. The time spent in the open arms and the total time spent in the closed arms were recorded using a video camera [43,44] and then statistically analyzed.

### 3.7. Collection of Brain Tissue Samples

All the animals were sacrificed by cervical decapitation under CO_2_ anesthesia on day 14, after the last dose of the drug was administered and the behavioral tests were performed. The entire brain was immediately resected and stored at −80 °C. The levels of neuroinflammatory biomarkers in brain tissue homogenate were determined as described in the following sections [45].

### 3.8. Enzyme-Linked Immunosorbent Assays

Brain samples were collected from the four groups (control, PIO, DOX, and DOX + PIO), subjected to sonication (Q-sonica homogenizer, 30 Hz pulses for 20 s) using neuronal lysis buffer (N-PER) (Thermo Scientific, Madison, WI, USA), and then centrifuged at 12,000× *g* for 10 min. The supernatant was collected, aliquoted into 200 μL vials, and stored at −80 °C. Protein present in the samples was quantified with the bicinchoninic acid assay (Pierce, Waltham, MA, USA), and the samples were subjected to enzyme-linked immunosorbent assay (ELISA) for TNF-α, IL-1β, and IL-6 as described in the manufacturer’s protocols (Mybiosource Inc., San Diego, CA, USA). The plates were read using the ELx800 Absorbance Microplate Reader (BioTek Instruments Inc., Winooski, VT, USA.) at a wavelength of 450 nm. The intensity of the color was compared with the standard and control to determine the concentrations of IL-1β, IL-6, and TNF-α in the samples. These data were then statistically analyzed.

### 3.9. Reverse-Transcription Quantitative Polymerase Chain Reaction

Total RNA was extracted from brain tissue samples (control, PIO, DOX, and DOX + PIO) by using the RNeasy kits (Qiagen, Hilden, Germany). Subsequently, total RNA was treated with RNase-free DNase (Ambion, Carlsbad, CA, USA) to eliminate possible traces of genomic DNA. The purity of RNA in each sample was determined by the NanoDrop ND-2000c spectrophotometer (Thermo Scientific, Labtech, UK). For RT-PCR, a specific primer for each gene (Table 1) was designed using Integrated DNA Technologies tools, and a 10 µM/μL working concentration was prepared. The ABScript II One-Step SYBR Green RT-qPCR Kit (RK20404: ABclonal Technology, Woburn, MA, USA) was used for reverse transcription and PCR quantification. Total RNA (400 ng) from each sample was reverse-transcribed to cDNA and run on the AiraMx Real-Time PCR system (Agilent Technologies, Santa Clara, CA, USA) according to the manufacturer’s instructions. A mixture of the PCR reaction SYBR Green RT-qPCR buffer, ABScript II Enzyme Mix, 10 μM of each forward and reverse primer, ROXII Reference Dye (50×), total RNA, and RNase-free H_2_O (up to 20 μL) was prepared. The thermal cycling conditions were as follows: reverse transcription, one cycle for 5 min at 42 °C; pre-denaturation, one cycle at 95 °C for 1 min; 40 reaction cycles each for 5 s at 95 °C; and 32–34 s at 60 °C. Samples were prepared in triplicate for three independent experiments to ensure that the results were valid. The data analysis was conducted automatically using the provided software (AiraMx 2.0 software) after adjusting the plate for comparative quantification. Gene expression levels were normalized to the expression of the GAPDH housekeeping gene. Transcription abundance for each gene relative to GAPDH transcription abundance was calculated and used to determine changes in the expression of each mRNA.

### 3.10. Statistical Analysis

All results were presented as mean ± standard error of the mean (SEM) and analyzed using the GraphPad Prism 9 software. The survival rate, changes in body weight, Y-maze, NOR, EPM, ELISA, and RT-PCR results for each group were analyzed using one-way ANOVA, followed by Tukey analysis. A *p*-value < 0.05 was considered to indicate statistical significance.

## 4. Discussion

The present study used DOX-treated experimental rats to model chemobrain and explore the effects of PIO on mitigating DOX-induced toxicity based on its effects on neuroinflammation and learning and memory behaviors.

The results of this study showed that two-week exposure to DOX (amounting to a total dose of 20 mg/kg, delivered intraperitoneally) was associated with general cognitive impairment. This was demonstrated through various behavioral tests. For example, the results of the Y-maze test showed that the animals that received only PIO could distinguish the novel arm from the starter or familiar arm and did not differ from the control group with regard to this ability. Thus, the administered PIO dose did not affect the rats’ ability to complete this task. However, rats treated with DOX showed some degree of cognitive impairment, as they spent less time in the novel arm and entered the novel arm fewer times than the control group rats. These cognitive deficits were improved when PIO was co-administered with DOX, as the rats that received both DOX and PIO performed better than those that received only DOX. Similarly, on the NOR task, the amount of time spent exploring the novel object was lower in the DOX group than in the control group. Further, the DOX-PIO-treated group spent more time exploring the novel object than the DXO-treated group. However, the results indicated that DOX-treated rats were not substantially impaired in memory function during this task. This may be due to the differential effect of DOX treatment on various brain regions, and not all brain regions are affected to the same degree. The current findings corroborated the previous reports showing that the NOR task is dependent on the ventral hippocampus [46]. Thus, DOX affects the ventral hippocampus to a lesser degree than it does other parts of the hippocampus.

In the EPM test, the DOX-treated rats showed a decrease in the total time spent in the open arms, and they had a significantly longer freezing time than rats from the other groups. These findings corroborated the previous reports showing anxiety-like behavior of elevated plus maze following DOX treatment, evidenced by freezing, defecation, and reduced motility in mice [47]. Anxiety is a natural response that promotes adaptive survival through escape from unnecessary danger. However, too much anxiety may disrupt regular brain functions, reducing the behavioral activity necessary for adaptation [48]. The amygdala plays a vital role in the expression of anxiety or fear, and the medial prefrontal cortex is important in the regulation of the amygdala-mediated expression of fear. Thus, the present findings may indicate the effect of DOX on the medial prefrontal cortex. The findings also demonstrate that the combination of PIO with DOX treatment had an ameliorative effect on the cognitive decline and anxiety-like behavior induced by DOX treatment. The results revealed that PIO and PIO + DOX treatments resulted in an increase in the amount of time spent in the open arm in the EPM test. Thus, PIO seems to have an ameliorative effect on the anxiety behavior caused by DOX. Our findings are consistent with a previous study reporting that pretreatment with PIO produced an improvement in anxiety and depressive behavior in rats induced by lipopolysaccharide injection [49]. Together, these results confirmed the cognitive impairment caused by DOX treatment and the improvement of these cognitive deficits by PIO co-administration. Overall, the results of these tests indicate that DOX caused significant memory impairment that was ameliorated by co-treatment with PIO.

Chemotherapy and chemotherapy-related neurotoxicity are associated with the release of proinflammatory cytokines. Further, neuropathic pain caused by the administration of chemotherapeutic agents is associated with a persistent increase in the expression of inflammatory cytokines in the brain [50]. Chemotherapy drugs, the nature of the tumor itself, and the patient’s long-term physiological and psychological stress can lead to an increase in the production of inflammatory cytokines such as TNF-α, IL-1, and IL-6. These cytokines have the ability to enter the brain through the BBB and cause a local inflammatory response in the brain, which could also lead to the destruction of the structure and integrity of the BBB [51]. In fact, doxorubicin causes severe neurotoxicity by stimulating the generation of inflammatory and pro-inflammatory mediators in the periphery, even though its ability to pass through the BBB is limited [52]. In agreement with these findings, our results also demonstrated the hyperactivation of neuroinflammatory mediators and a statistically significant increase of IL-1β, IL-6, and TNF-α following DOX treatment as compared with the control animals. Moreover, the elevation in neuroinflammatory cytokines in DOX-treated rats was associated with cognitive impairment. PIO has anti-inflammatory [26,27] and antioxidant properties [28] that might ameliorate chemotherapy toxicity. Accordingly, in vivo studies have found that PPAR-γ agonists have the ability to inhibit β-amyloid-stimulated expression of IL-6 and TNFα [53] and that PIO improves visuospatial and long-term memory [54]. In line with these findings, co-administration of DOX and PIO in the present study resulted in the alleviation of cognitive impairment. In order to confirm the results, we performed RT-PCR to quantify the mRNA expression levels of IL-6 and TNF-α in the brain tissue of the experimental animals, and the results demonstrated a consistent elevation in DOX-treated rats that was reversed with the co-administration of PIO. These findings confirm the neuroprotective effect of PIO that has been reported in previous studies.

## 5. Conclusions

The present findings prove our initially proposed hypothesis that DOX induces cognitive dysfunction through an increase in the levels of pro-inflammatory cytokines and that PIO has a neuroprotective effect against the cognitive impairment and elevation of pro-inflammatory cytokines caused by DOX treatment. Thus, PIO may have potential for the treatment of chemobrain in the future. Future investigations into the detailed mechanisms of DOX-induced cognitive impairment at different time points after treatment and the neuroprotective mechanisms of PIO would help in the development of PIO-based treatment strategies as well as the identification of other therapeutic targets and agents.

## Figures and Tables

**Figure 1 molecules-28-04775-f001:**
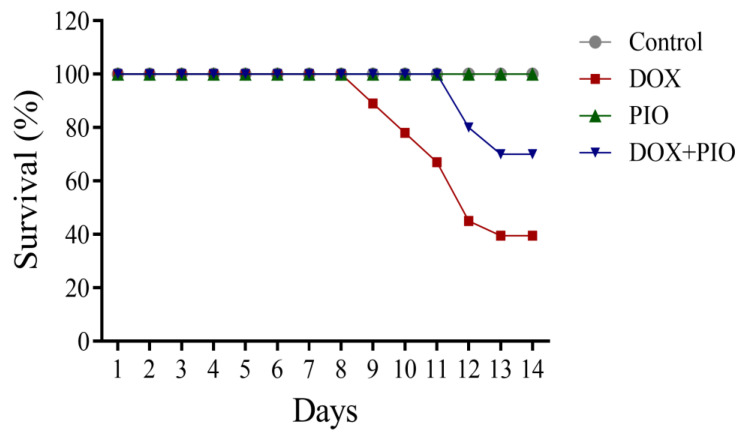
Effects of DOX and PIO on survival. Data represented as mean ± SEM (*n* = 10 for each group).

**Figure 2 molecules-28-04775-f002:**
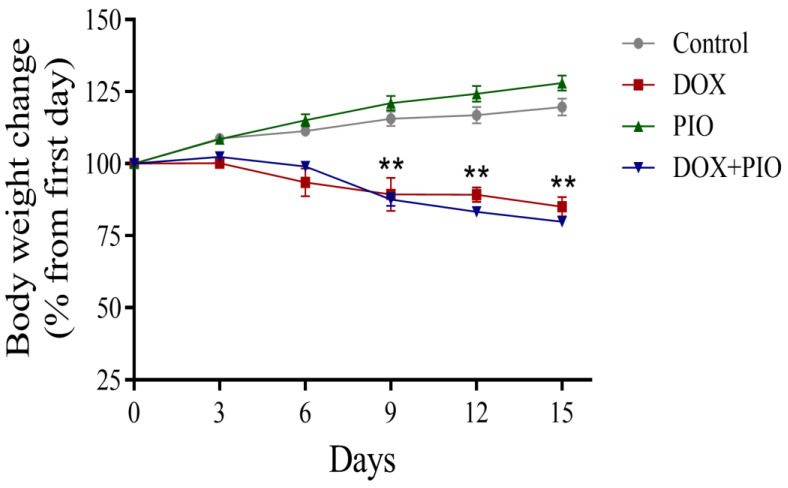
Effects of DOX and PIO on body weight. Data represent mean ± SEM and ** *p* < 0.01 was considered significant by ANOVA followed by Tukey analysis as compared to corresponding weight on first day.

**Figure 3 molecules-28-04775-f003:**
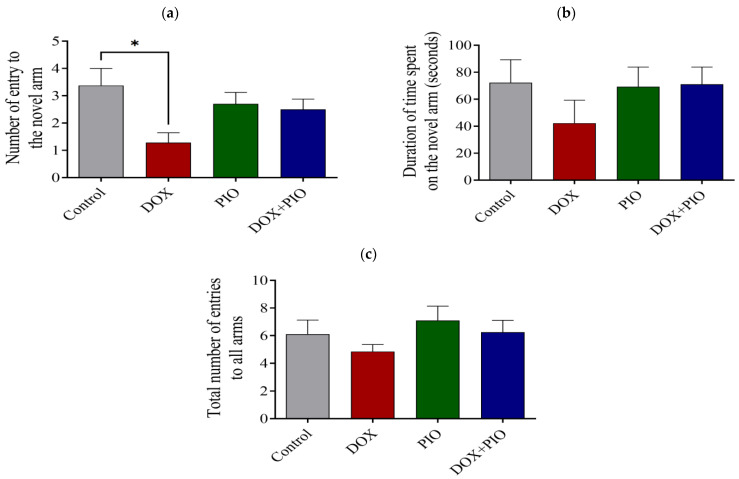
Effect of PIO on DOX-induced behavioral changes on Y-maze test in rats. (**a**) Effects of DOX and PIO on the number of entries into the novel arm in the Y-maze. (**b**) Effects of DOX and PIO on the total time spent in the novel arm in the Y-maze test. (**c**) Total number of entries into all arms (familiar and novel). Data represented as mean ± SEM and analyzed by ANOVA followed by Tukey test. * *p* < 0.05 was considered significant when compared with control.

**Figure 4 molecules-28-04775-f004:**
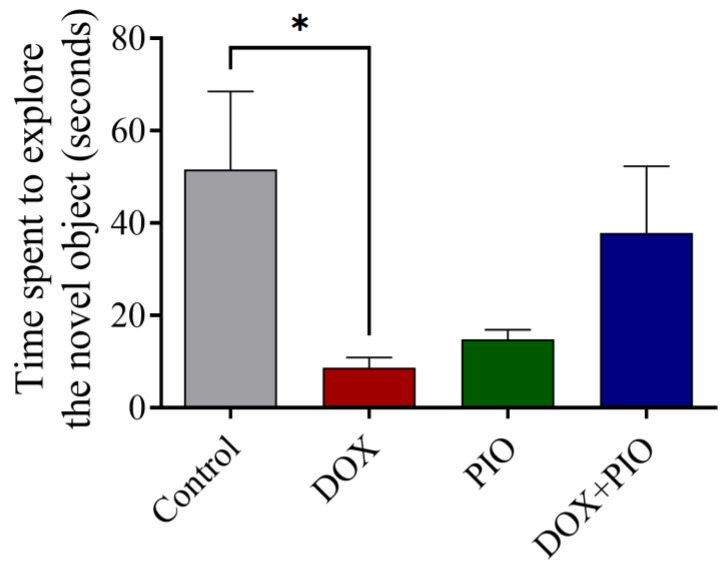
Effect of PIO treatment on DOX-induced changes in the behavior of rats on novel object recognition test. Data represented as mean ± SEM and analyzed by ANOVA followed by Tukey test. * *p* < 0.05 was considered significant when compared with control.

**Figure 5 molecules-28-04775-f005:**
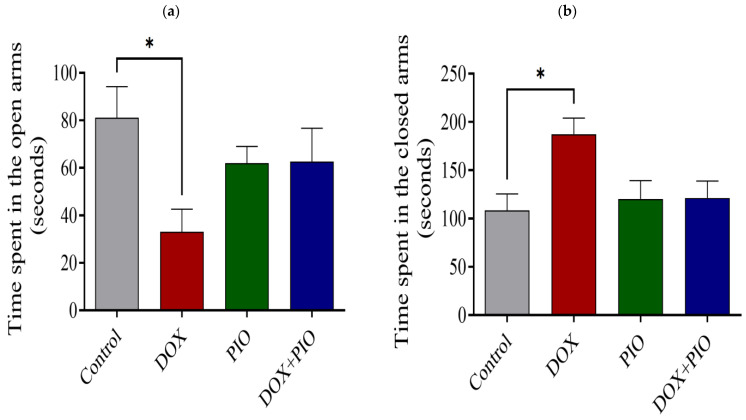
Effect of PIO on DOX-induced behavioral changes in rats on EPM test. (**a**) Time spent in the closed arms. (**b**) Time spent in the open arms. Data represented as mean ± SEM and analyzed by ANOVA followed by Tukey test. * *p* < 0.05 was considered significant when compared with control.

**Figure 6 molecules-28-04775-f006:**
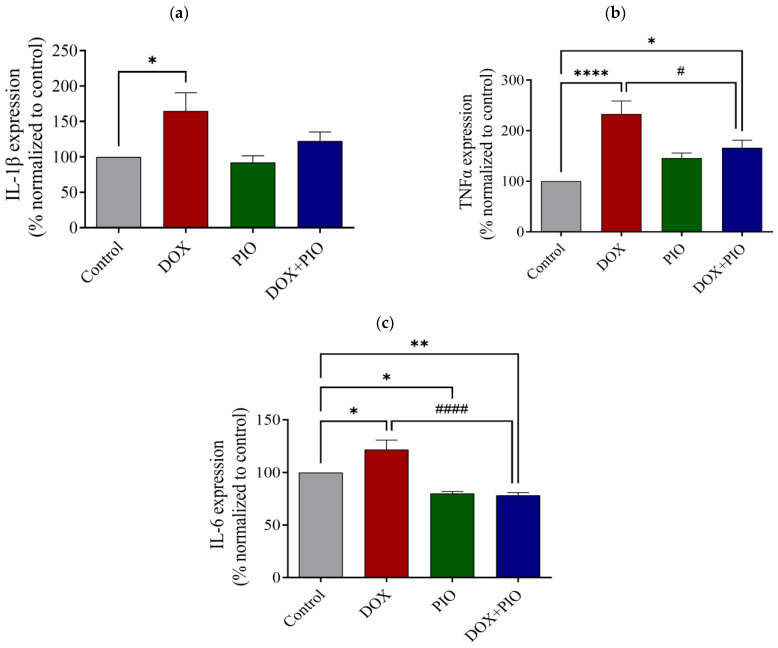
Effect of DOX and PIO on IL-1β (**a**), TNF-α (**b**), and IL-6 (**c**) levels in rat brain. Data represented as mean ± SEM and analyzed by ANOVA followed by Tukey test and *p* < 0.05 was considered significant. * *p*< 0.05, ** *p* < 0.01, **** *p* < 0.0001 when compared with control and # *p* < 0.05, #### *p* < 0.0001 when compared with DOX.

**Table 1 molecules-28-04775-t001:** The specific primer pairs used in polymerase chain reaction.

Gene	Primer Sequence (5′–3′)	Length/bp
TNF-α FWD	ACCTTATCTACTCCCAGGTTCT	87
TNF-α REV	GGCTGACTTTCTCCTGGTATG	
IL6 FWD	GCCAGAGTCATTCAGAGCAATA	87
IL6 REV	TTAGGAGAGCATTGGAAGTTGG	
GAPDH FWD	ACTCCCATTCTTCCACCTTTG	104
GAPDH REV	CCCTGTTGCTGTAGCCATATT	

## Data Availability

Available upon request for reasonable reason.

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
