# Peer review of "The Ameliorative Effect of Pioglitazone against Neuroinflammation Caused by Doxorubicin in Rats"

_molecules, 2023, doi:10.3390/molecules28124775_

Round 1

Reviewer 1 Report

Comments to authors

The manuscript by Alsaud et al. investigated the effect of pioglitazone on doxorubicin-induced chemobrain in female rats. In this work, the authors studied few behavioral tests and ELISA, and RT-PCR analysis of inflammatory factors in rat brain. The idea of the planned study is excellent and can be of reader’s interest. However, the manuscript has several serious issues that are drastically hampering the quality of the manuscript.

Major Comments

1.     The title of the manuscript reads “Neuroprotective Effect of Pioglitazone Against Neuroinflammation Caused by Doxorubicin in Rats” while no any parameter has been performed that indicates neuroprotection. Authors are suggested to perform any of the assays that indicate neuroprotection (e.g., Cresyl violet staining, TUNEL assay, IHC of neuronal markers like NeuN).

2.     The study was executed on female rats only. The reason for using them has not been explored in any part of the manuscript. The same should be clarified. Which strain of the rats selected should also be mentioned?  The source of animal procurement and housing should be clearly mentioned under the heading “Animals”.

3.     Doses of DOX and PIO; how doses of DOX were prepared should be clearly mentioned as nothing is given about the vehicle in which DOX was dissolved. Moreover, mention the doses of PIO as only concentration of PIO has been mentioned.

4.     Was total RNA isolated by two different procedures (by TRIzol reagent and RNeasy kits)? If so, then why? Otherwise correct accordingly.

5.     The result part is poorly written. There are four different groups namely; Control, DOX, PIO, and DOX+PIO and the results for each parameter should be written firstly by comparing DOX with Control (DOX vs Control), then DOX+PIO by DOX (DOX+PIO vs DOX). The comparing and writing results among other groups do not make any sense in the context of this study. As the goal of this study is to determine the detrimental effect of DOX on the brain and then investigate the beneficial effect of PIO. Authors can also mention the results between control vs PIO (to represent the effect of PIO in normal conditions) that showed non-significant changes for most of the studied parameters.  

6.     Figure 1 represents the day-wise survival rate of rats from all experimental groups but this is not giving a clear picture of the results written in results section. So, I suggest presenting the data in the form total number of rats that survived at the end of the treatment period as well.

7.     Authors stated that the body weight of rats was determined during two weeks of the treatment periods while Figure 2. shows the data of only 9 days. Moreover, the trendline plot suggests that body weight was measured on the 3rd 5th and 8th days of the treatment period. Why there is a difference in the time points (0 to 3rd; after two days, 3rd to 5th; after one day and 5th to 8th; after two days) for body weight measurement? this should be clarified. Based on the data provided in Figure 2, how one can determine the change in body weight during the course of treatment? Hence, the authors are suggested to calculate the gain in body weight (subtract the initial body weight from the final body weight) and present it as Mean + SEM.   

8.     Authors are suggested to carefully check the data for all the parameters as in most of the cases results written in the results section is not matching with their respective figures. Also suggested rigorously checking the data, and statistical significance level and using different symbols for showing significance (e.g., you can use “*” for DOX vs Control and “#” for DOX+PIO vs DOX for better understanding).

9.     All the figures need to be corrected as the significance level should be mentioned for DOX vs Control and DOX+PIO vs DOX only, to minimize confusion to readers. Also, the size of all the barographs is different, thus authors can take serious note of the correction of all the figures. I also assume that the authors have tried their best to maximize the number of figures. Figures No 6, 7, and 8 can be merged to make one Figure and similarly, Figure No 9 and 10 can be merged.

10.  Discussion part of the manuscript is also written poorly. Does discuss the results of the PIO vs control group have any relevance with respect to your interpretation? (Line No 344 to 347).

11.  How many rats were used to carry out every parameter should be clearly mentioned in the methodology section and in the respective figure legends (including details of ANOVA). For assistance you can consult these articles; https://doi.org/10.3390/brainsci13040563, https://doi.org/10.3390/molecules27238294 etc.

Minor comments

1.     ELISA and RT-PCR results for DOX+PIO are missing in the abstract.

2.     Words like TNF-a and IL-1b should be written correctly as TNF-α and IL-1β by inserting symbols instead of alphabets.

3.     Why IL-1β was not assayed by RT-PCR analysis? While assayed by ELISA.

4.     RT-PCR is not included in the statistical analysis statement.

5.     Most of places “PIO” is written in lowercase as “Pio” (including in figures).

Overall, I recommend an in-depth revision of the manuscript including the addition of new assays suggesting neuroprotection.      

Author Response

Reviewer #1

The manuscript by Alsaud et al. investigated the effect of pioglitazone on doxorubicin-induced chemobrain in female rats. In this work, the authors studied few behavioral tests and ELISA, and RT-PCR analysis of inflammatory factors in rat brain. The idea of the planned study is excellent and can be of reader’s interest. However, the manuscript has several serious issues that are drastically hampering the quality of the manuscript.

Reply:                      Thank you very much for your comment and based on your comment the article has been thoroughly revised.

Major Comments

  1. The title of the manuscript reads “Neuroprotective Effect of Pioglitazone Against Neuroinflammation Caused by Doxorubicin in Rats” while no any parameter has been performed that indicates neuroprotection. Authors are suggested to perform any of the assays that indicate neuroprotection (e.g., Cresyl violet staining, TUNEL assay, IHC of neuronal markers like NeuN).

Reply:             Although IHC and TUNEL assay are considered as more suitable test for neuroprotection but due to limitation of lab facilities, we couldn’t perform these tests. However, there are many research reporting cytokines as markers of neurotoxicity. Further, many more studies have been published with such parameters for the neuroprotective effect.

  1. Khadrawy YA, Hosny EN, Mohammed HS. Protective effect of nanocurcumin against neurotoxicity induced by doxorubicin in rat’s brain. NeuroToxicology. 2021;85:1-9.
  2. Chao CC, Hu SX, Ehrlich L, Peterson PK. Interleukin-1 and tumor necrosis factor-α synergistically mediate neurotoxicity: involvement of nitric oxide and of N-methyl-D-aspartate receptors. Brain, behavior, and immunity. 1995 Dec 1;9(4):355-65.
  3. Rauf A, Badoni H, Abu-Izneid T, Olatunde A, Rahman MM, Painuli S, Semwal P, Wilairatana P, Mubarak MS. Neuroinflammatory markers: key indicators in the pathology of neurodegenerative diseases. Molecules. 2022 May 17;27(10):3194.
  4. Piao HZ, Jin SA, Chun HS, Lee JC, Kim WK. Neuroprotective effect of wogonin: Potential roles of inflammatory cytokines. Arch Pharm Res. 2004; 27:930-936.

  1. The study was executed on female rats only. The reason for using them has not been explored in any part of the manuscript. The same should be clarified. Which strain of the rats selected should also be mentioned?  The source of animal procurement and housing should be clearly mentioned under the heading “Animals”.

Reply:             The reason for using female rats has been included in the “introduction” section. The source of animal procurement and housing condition has been revised in “materials and method” section as suggested.

  1. Doses of DOX and PIO; how doses of DOX were prepared should be clearly mentioned as nothing is given about the vehicle in which DOX was dissolved. Moreover, mention the doses of PIO as only concentration of PIO has been mentioned.

Reply:             The “methodology” has been modified as suggested.

  1. Was total RNA isolated by two different procedures (by TRIzol reagent and RNeasy kits)? If so, then why? Otherwise correct accordingly.

Reply:             Typographical error has been corrected.

  1. The result part is poorly written. There are four different groups namely; Control, DOX, PIO, and DOX+PIO and the results for each parameter should be written firstly by comparing DOX with Control (DOX vs Control), then DOX+PIO by DOX (DOX+PIO vs DOX). The comparing and writing results among other groups do not make any sense in the context of this study. As the goal of this study is to determine the detrimental effect of DOX on the brain and then investigate the beneficial effect of PIO. Authors can also mention the results between control vs PIO (to represent the effect of PIO in normal conditions) that showed non-significant changes for most of the studied parameters.  

Reply:             The result section has been re-written as suggested.

  1. Figure 1 represents the day-wise survival rate of rats from all experimental groups but this is not giving a clear picture of the results written in results section. So, I suggest presenting the data in the form total number of rats that survived at the end of the treatment period as well.

Reply:             Modified as suggested.

  1. Authors stated that the body weight of rats was determined during two weeks of the treatment periods while Figure 2. shows the data of only 9 days. Moreover, the trendline plot suggests that body weight was measured on the 3rd5th and 8th days of the treatment period. Why there is a difference in the time points (0 to 3rd; after two days, 3rd to 5th; after one day and 5th to 8th; after two days) for body weight measurement? this should be clarified. Based on the data provided in Figure 2, how one can determine the change in body weight during the course of treatment? Hence, the authors are suggested to calculate the gain in body weight (subtract the initial body weight from the final body weight) and present it as Mean + SEM.

Reply:             Agree with the reviewer’s suggestion. However, in the current study all the treatment group didn’t exhibited gain in body weigh except PIO. Therefore, we calculated change in body weight which has been adopted by several previous publications.

  1. Wong J, Tran TL, Lynch KA, Wood LJ. Dexamethasone exacerbates cytotoxic chemotherapy induced lethargy and weight loss in female tumor free mice. Cancer Biol Ther. 2018;19:87-96.
  2. Mohammed AI, Celentano A, Paolini R, Low JT, McCullough MJ, O’Reilly LA, Cirillo N. Characterization of a novel dual murine model of chemotherapy-induced oral and intestincal mucositis. Scientific Reports 2023;1396.
  3. Kumar SA, Needham RJ, Abraham K, Bridgewater HE, Garbutt LA, Xandri-Monje H, Dallmann R, Perrier S, Sadler PJ, Levi F. Dose- and time-dependent tolerability and efficacy of organo-osmium complex FY26 and its tissue pharmacokinetics in hepatocarcinoma-bearing mice. Metallomics. 2021;13((2):003
  4. Ghosh AR, Alsayari A, Habib AH, Wahab S, Nadig APR, Rafeeq MM, Binothman N, Aljadani M, Al-Dhuayan IS, Alaqeel NK, Khalid M, Krishna KL. Anti-tumor potential of Gymnema sylvestre saponin rich fraction on in vitro breast cancer cell lines and in vivo tumor-bearing mouse models. Antioxidants 2023;12:134.
  5. Lu J, Lou Y, Zhang Y, Zhong R, Zhang W, Zhang X, Wang H, Chu T, Han B, Zhong H. Paclitaxel has a reduced toxicity profile in healthy rats after polymeric micellar nanoparticle delivery. International Journal of Nanomedicine. 2023;263-76.

  1. Authors are suggested to carefully check the data for all the parameters as in most of the cases results written in the results section is not matching with their respective figures. Also suggested rigorously checking the data, and statistical significance level and using different symbols for showing significance (e.g., you can use “*” for DOX vs Control and “#” for DOX+PIO vs DOX for better understanding).

Reply:             Checked and modified as suggested.

  1. All the figures need to be corrected as the significance level should be mentioned for DOX vs Control and DOX+PIO vs DOX only, to minimize confusion to readers. Also, the size of all the barographs is different, thus authors can take serious note of the correction of all the figures. I also assume that the authors have tried their best to maximize the number of figures. Figures No 6, 7, and 8 can be merged to make one Figure and similarly, Figure No 9 and 10 can be merged.

Reply:             Modified as suggested.

  1. Discussion part of the manuscript is also written poorly. Does discuss the results of the PIO vs control group have any relevance with respect to your interpretation? (Line No 344 to 347).

Reply:             Discussion section has been revised as suggested.

  1. How many rats were used to carry out every parameter should be clearly mentioned in the methodology section and in the respective figure legends (including details of ANOVA). For assistance you can consult these articles; https://doi.org/10.3390/brainsci13040563https://doi.org/10.3390/molecules27238294

Reply:             Modified as suggested.

Major Comments

  1. ELISA and RT-PCR results for DOX+PIO are missing in the abstract.

Reply:             Included in the abstract section

  1. Words like TNF-a and IL-1b should be written correctly as TNF-α and IL-1β by inserting symbols instead of alphabets.

Reply:             Modified as suggested

  1. Why IL-1β was not assayed by RT-PCR analysis? While assayed by ELISA.

Reply:             Included in the abstract section

  1. RT-PCR is not included in the statistical analysis statement.

Reply:             Included in the statistical analysis section as suggested.

  1. Most of places “PIO” is written in lowercase as “Pio” (including in figures).

Reply:             Modified as suggested.

Reviewer 2 Report

Authors have reported Neuroprotective Effect of Pioglitazone Against Neuroinflammation Caused by Doxorubicin in Rats. Overall, the manuscript is well-written and well-conceived. Abstract provides an overview of the research conducted. The introduction section is sufficient with enough literature background to justify the novelty of the work. The methodology section is sound and reproduceable. Results are well presented dully supported by the figures. Discussion section is appropriate and justify the results obtained. Finally, conclusion section appropriately drew main findings of the research. In my opinion, the paper may be accepted in its current form.

Author Response

Reviewer #2

Authors have reported Neuroprotective Effect of Pioglitazone Against Neuroinflammation Caused by Doxorubicin in Rats. Overall, the manuscript is well-written and well-conceived. Abstract provides an overview of the research conducted. The introduction section is sufficient with enough literature background to justify the novelty of the work. The methodology section is sound and reproduceable. Results are well presented dully supported by the figures. Discussion section is appropriate and justify the results obtained. Finally, conclusion section appropriately drew main findings of the research. In my opinion, the paper may be accepted in its current form.

Reply:             Thank you very much for your comments.

Reviewer 3 Report

Thank you very much for allowing me to review the article entitled "Neuroprotective Effect of Pioglitazone Against Neuroinflammation Caused by Doxorubicin in Rats" (molecules-2435006), which has been submitted to the "Medicinal Chemistry" section in the Special Issue "Bioactive Compounds for Brain Ischemia and Neurodegenerative Disease."

The aim of this study is to determine the protective effect of pioglitazone (PIO), an antidiabetic agent, against cognitive impairment induced by doxorubicin (DOX). At the end of the study, it is indicated that DOX treatment resulted in memory function impairment in rat models by inducing neuroinflammation, and that PIO treatment alleviated the cognitive effects of DOX.

The introduction is well presented, and the references are relevant to the current knowledge on the subject. The hypothesis and objective are well stated.

Materials and methods: Why have only female rats been used?. The design and analysis strategy are appropriate for the stated objective.

Results: They have been structured to facilitate understanding of each section. The graphs and tables greatly enhance the comprehension of the study.

Discussion: It is well presented, but I would recommend including the limitations of the study that should be considered for future research, as well as the next steps in this line of investigation.

The conclusion is a result of the study.

Author Response

Reviewer #3

  1. Thank you very much for allowing me to review the article entitled "Neuroprotective Effect of Pioglitazone Against Neuroinflammation Caused by Doxorubicin in Rats" (molecules-2435006), which has been submitted to the "Medicinal Chemistry" section in the Special Issue "Bioactive Compounds for Brain Ischemia and Neurodegenerative Disease."

Reply: Thank you very much for your comments.

  1. The aim of this study is to determine the protective effect of pioglitazone (PIO), an antidiabetic agent, against cognitive impairment induced by doxorubicin (DOX). At the end of the study, it is indicated that DOX treatment resulted in memory function impairment in rat models by inducing neuroinflammation, and that PIO treatment alleviated the cognitive effects of DOX.

Reply: Thank you very much for your comments.

  1. The introduction is well presented, and the references are relevant to the current knowledge on the subject. The hypothesis and objective are well stated.

Reply: Thank you very much for your comments.

  1. Materials and methods: Why have only female rats been used?. The design and analysis strategy are appropriate for the stated objective.

Reply: The reason for using female rats has been included in the “introduction” section.

  1. Results: They have been structured to facilitate understanding of each section. The graphs and tables greatly enhance the comprehension of the study.

Reply: Thank you very much for your comments.

  1. Discussion: It is well presented, but I would recommend including the limitations of the study that should be considered for future research, as well as the next steps in this line of investigation.

Reply: Thank you very much for your comments.

  1. The conclusion is a result of the study.

Reply: Thank you very much for your comments.

Round 2

Reviewer 1 Report

The authors have significantly improved the manuscript and clarified most of the queries raised by me. The only shortcomings in the manuscript that still needs to be rectified are as below. 

1. In response to my major query 1, the authors tried to justify that inflammation markers can be applied to assays neuroprotection. In this manuscript, none of the parameters that indicate neuroprotection have been studied, so based on this data neuroprotective role of Pioglitazone can not be established. Authors are suggested to modify the word "neuroprotective" in the title and manuscript to "ameliorative". 

2. “PIO” is still written as “Pio” in a few instances.  

Author Response

Response to the reviewer’s comments_R2

I would like to extend my sincere thanks to the editor and all reviewers for their comments and suggestions. The manuscript has been revised as per their advice and has been highlighted in “red” in the manuscript. I believe that addressing all the comments of the reviewers have substantially improved the quality of the manuscript. Detailed point to point replies to the comments are given below.

Reviewer #1

The authors have significantly improved the manuscript and clarified most of the queries raised by me. The only shortcomings in the manuscript that still needs to be rectified are as below. 

Reply:                      Thank you very much for your comment.

  1. In response to my major query 1, the authors tried to justify that inflammation markers can be applied to assays neuroprotection. In this manuscript, none of the parameters that indicate neuroprotection have been studied, so based on this data neuroprotective role of Pioglitazone cannot be established. Authors are suggested to modify the word "neuroprotective" in the title and manuscript to "ameliorative".

Reply:             The title has been modified as suggested.

  1. “PIO” is still written as “Pio” in a few instances.

Reply:             Typographical errors in figures 2 and 3 have been corrected as per suggestion.
